# Learning Relative Gene Expression Trends from Pathology Images in Spatial Transcriptomics

Kazuya Nishimura[1]    Haruka Hirose[1]    Ryoma Bise[2]    Kaito Shiku[2]    Yasuhiro Kojima[1]

[1] Laboratory of Computational Life Science, National Cancer Center Japan
[2] Department of Advanced Information Technology, Kyushu University, Japan
kanishi4@ncc.go.jp

## Abstract

Gene expression estimation from pathology images has the potential to reduce the RNA sequencing cost. Point-wise loss functions have been widely used to minimize the discrepancy between predicted and absolute gene expression values. However, due to the complexity of the sequencing techniques and intrinsic variability across cells, the observed gene expression contains stochastic noise and batch effects, and estimating the absolute expression values accurately remains a significant challenge. To mitigate this, we propose a novel objective of learning relative expression patterns rather than absolute levels. We assume that the relative expression levels of genes exhibit consistent patterns across independent experiments, even when absolute expression values are affected by batch effects and stochastic noise in tissue samples. Based on the assumption, we model the relation and propose a novel loss function called STRank that is robust to noise and batch effects. Experiments using synthetic datasets and real datasets demonstrate the effectiveness of the proposed method. The code is available at `https://github.com/naivete5656/STRank`.

## 1 Introduction

With the development of spatial transcriptomic techniques (ST), the comprehensive gene expression profile can be captured on a small spot with a spatial location corresponding to the pathology image [16]. Due to the high cost of acquiring spatial transcriptomics (ST) data, there is growing interest in using computer vision techniques to estimate gene expression from pathology images as a more affordable way [19, 30, 4, 7].

One of the main difficulties in estimating gene expression from pathology images is the batch effects and stochastic fluctuations in observed data. As shown in Figure 1 (a), differences in reagent batches, equipment, and other technical factors in the measurement process (i.e., batch effects) can cause variations in data scaling across tissues [11, 18]. Additionally, due to cellular heterogeneity and temporal dynamics, the observed gene expression level stochastically fluctuates even though the appearance of the pathology image is the same as shown in Figure 1 (b).

Although mean squared error (MSE) is commonly used in the previous estimation methods [7, 19, 30, 4], it is hard to capture variations from the data containing batch effects and stochastic noise. MSE loss focuses on predicting the absolute values of gene expression without correcting for batch effects. Consequently, models trained with MSE loss may inadvertently learn patient-specific biases rather than biologically relevant signals. Additionally, since MSE loss does not model stochastic noise explicitly, it can not account for the significance of biological signals from expression count data.

In this paper, we aim to estimate the relative expression relation instead of directly estimating absolute values of gene expression. The key hypothesis of this paper is that relative gene expression trends

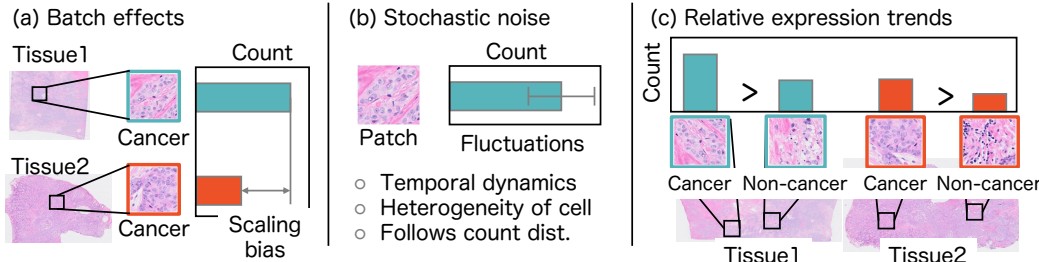

Figure 1: (a) Illustration of scaling bias due to batch effects, (b) stochastic noise, and (c) our hypothesis: learning relative expression trends. Even in the presence of batch effects and stochastic noise, the relative expression trends between patches are preserved.

between image patches are preserved, even when batch effects or stochastic noise are contained in count values. For instance, as illustrated in Figure 1 (c), if we extract cancerous and non-cancerous patches from several tissues, the expression of the cancer cell-specific gene is expected to exhibit higher expression in cancerous patches for all the tissues. Although the absolute expression values and their scales may vary across tissues, we posit that the relative relation of expression between patches remains consistent. In addition, the relative expression relation has been widely used for downstream analysis, such as differential expression analysis, which detects the relative expression difference between clusters. Therefore, capturing relative expression differences between patches within each tissue is more reasonable than directly estimating the absolute value of gene expression.

Learning to rank (i.e., ranking loss) is one of the solutions to learn the relationships between samples [3, 22, 9, 10, 27]. This ranking loss learns which of a given pair of samples has a higher score, and the pairwise learning approach can mitigate batch effects. However, because signal-to-noise ratios of lowly expressed genes tend to be lower than those of highly expressed genes, stochastic noise can alter the relative ranking between samples when gene expression levels are low. Therefore, it is essential to model the relative relationships in a manner that reflects the probabilistic characteristics of gene expression data.

To address the challenge of learning relative relationships in gene expression data affected by stochastic noise, we propose a novel loss function that models gene expression as a discrete probabilistic distribution conditioned on pairwise or listwise input. Specifically, we assume that the expression counts of paired patches are the consequence of the counting process given relative frequencies among the patches; we assume a binomial distribution for pairwise and a multinomial distribution for listwise scenarios. This formulation enables the model to capture relative relationships in a manner that is consistent with the probabilistic nature of observed count data.

To confirm the characteristics of our proposed loss function, we compared a previous loss function with ours using a synthetic dataset. The experiments demonstrated the effectiveness of batch effects and stochastic noise in the low signal situation. Moreover, experiments using real datasets show the generability of our loss function.

Our contributions are highlighted as follows:

- We redefine the task setup of gene expression estimation from pathology images as a rank score estimation setting. The setting is more practical and realistic for the downstream tasks of using gene expression.

- We propose a noise-robust loss function designed to handle batch effects and probabilistic noise, which dynamically adjusts its weighting based on the relative magnitude of expression for spatial transcriptomic data (STRank). This allows the model to learn more effectively, even when the expressions are sparse.

- We demonstrate the effectiveness of our loss function for batch effect and stochastic noise using synthetic datasets. In addition, we validated the robustness of the proposed method in expression estimation using real-world data, confirming its effectiveness under practical conditions.

## 2 Setup and Notation

Let $\mathcal{X}^{(n)} = \{x^{n,i}\}_{i=1}^{N^p}$ denote a set of image patches in $n$-th tissue ($n = 1, \ldots, N^{(\text{tissue})}$) and $\mathcal{E}^{(n)} = \{e^{n,i}\}_{i=1}^{N^p}$ be a set of their corresponding gene expression levels at each patch, where $N^p$ is the number of patches. The $e^i$ is an $N^g$-dimensional vector and indicates gene expression level for each gene, where $N^g$ is the number of genes. Unlike previous expression estimation setups that directly estimate expression value $e^{n,i}$ from $x^{n,i}$ without considering tissue $n$, we aim to learn a function $f : x^{n,i} \to r^{n,i}$ that estimates the rank score [3] for each gene $g$, which reflects the relative relation of gene expression between the given patches from the same tissue. The $r_g^{n,i}$ is the scale-invariant, it reflects relative relation among expression of tissue $n$, where if $e_g^{n,i} > e_g^{n,j}$, then the rank score should be $r_g^{n,i} > r_g^{n,j}$. Since raw gene expression values have some scaling biases introduced by the complexity of the observation technique and experimental conditions, our setup is more intuitive and practical than directly estimating the raw expression value. For readability, we omit the subscript $n$ in, $x^{n,i}$ and $e^{n,i}$. These values are referred to as $x^i$ and $e^i$, which are samples in tissue $n$.

To motivate this approach, we first summarise the two losses most widely adopted in prior work — MSE (pointwise) and Rank (pairwise) — and clarify why they remain vulnerable to either batch effects or stochastic noise.

A mean squared error loss between raw expression value $e^i$ and the estimated expression has been widely used for gene expression estimation from pathology image [7, 19, 30]. The mean squared error (MSE) loss for a sample $x^i$ is defined as:

$$L_{\text{MSE}}(e^i, \hat{r}^i) = \frac{1}{N^g} \left|\left| e^i - \hat{r}^i \right|\right|^2, \tag{1}$$

where $|| \cdot ||$ is the L2 norm, and $\hat{r}^i$ is the output of a function $f$ that estimates the expression value, such as a neural network, $\hat{r}^i = f(x^i)$.

The loss function is calculated based on one patch (*i.e.,* pointwise input). If there are batch effects between $\mathcal{E}^{(n)}$ and $\mathcal{E}^{(m)}$, the loss function can be influenced by scaling bias due to batch effects because the loss does not consider relations between samples obtained from different tissues. When patient data is imbalanced, the model may disproportionately rely on data from certain individuals. This can cause the model to learn spurious patterns, resulting in experimental bias.

To deal with the batch effects in the target data, the pairwise loss, such as Rank loss [3] or listwise loss, such as PCC loss [25, 2] that learn relative relations from multiple inputs, is one of the countermeasures. Ranking loss is designed to capture the relative ordering between pairwise samples. Given pairwise samples from the same tissue $n$, $x^i$ and $x^j$, such that their gene expression satisfies $e^i > e^j$, the loss is computed for each pairwise sample in the batch as follows:

$$\mathcal{L}_{\text{Rank}}(\hat{r}^i, \hat{r}^j) = \max\left(0, \hat{r}_g^i - \hat{r}_g^j + \varepsilon\right), \tag{2}$$

where $\varepsilon$ is a margin value.

While we could learn pairwise relations in tissue by introducing the Rank loss, since the Rank loss does not consider stochastic fluctuation, it is difficult to capture the signal in low signal conditions. The gene expression profile captured by spatial transcriptomics is very sparse and noisy. Therefore, to capture the signal from such a sparse dataset, we should consider the probabilistic model for the loss function.

## 3 Spatial Transcriptomics Ranking Loss

The motivation of our Spatial Transcriptomics Ranking Loss (STRank) is to learn the relative relationships of gene expression by considering the stochastic noise effect using the distribution of count data by modeling gene expression counts at multiple spots. Similar to the learning to rank setting, we consider two setups: pairwise and listwise loss functions.

### 3.1 Pairwise STRank Loss

Let us consider the pairwise situation similar to learning to rank [3]. Given a pair of patches $i, j$ obtained from a single tissue, we train the model $f$ to predict the rank score $r^i$, which reflects

the relation of gene expression $e^i, e^j$ between the samples from patch images $x^i, x^j$. In contrast to the conventional rank loss function [3] that focuses only on learning ordinal relationships (i.e., which sample is larger), our proposed loss function captures relative differences by incorporating the magnitude of gene expression level.

We assume that the expression count $e^i$ on the spot $i$, given the pairwise patches $x^i, x^j$ and total expression $\mathbf{t}^{i,j}$, follows a Binomial distribution.

$$\Pr(e^i|x^i, x^j, t^{i,j}) = \prod_{g=1}^{N^g} \Pr(e_g^i|x^i, x^j, t_g^{i,j}), \ \Pr(e_g^i|x^i, x^j, t_g^{i,j}) = \text{Binomial}(t_g^{i,j}, p_g^i), \quad (3)$$

where $N^g$ is the number of genes, $p_g^i$ is the frequency parameter of Bionomial distribution, which quantify how frequently the gene $g$ is observed at spot $i$ given the expression count is derived from either spot $i$ or $j$, and $t_g^{i,j}$ is the total expression level of gene $g$ across these two spots: $t_g^{i,j} = e_g^i + e_g^j$. This modeling approach accounts for the unique statistical characteristics of count data. It enables adaptive weighting of inter-sample relationships based on the observed count levels, thereby improving the ability to learn from count data distributions which has stochastic fluctuations.

Given pairwise patches, $x^i$ and $x^j$, the model $f$ output scores, $\hat{r}^i$ and $\hat{r}^j$, where $\hat{r}^i, \hat{r}^j \in \mathbb{R}^{N^g}$. A softmax function is then applied between $\hat{r}^i$ and $\hat{r}^j$.

$$\hat{p}_g^i = \frac{\exp(\hat{r}_g^i)}{exp(\hat{r}_g^i) + exp(\hat{r}_g^j)}, \quad \hat{p}_g^j = \frac{\exp(\hat{r}_g^j)}{exp(\hat{r}_g^j) + exp(\hat{r}_g^i)}, \quad \hat{p}_g^j = 1 - \hat{p}_g^i. \quad (4)$$

Our loss function models the predicted probabilities $p_g^i$ and $p_g^j$ as parameters of a binomial distribution, and the model is trained by minimizing the negative log-likelihood of the binomial distribution, thereby aligning the predicted distributions with the observed count-based outcomes. The negative log-likelihood can be decomposed into the following form:

$$-\log \Pr(e^i|x^i, x^j, t^{i,j}) = -\log \left( \prod_{g=1}^{N^g} \binom{t_g^{i,j}}{e_g^i} p_g^{i\,e_g^i} p_g^{j\,e_g^j} \right) \quad (5)$$

$$= -\sum_{g=1}^{N^g} \left( e_g^i \log p_g^i + e_g^j \log p_g^j + \log \binom{t_g^{i,j}}{e_g^i} \right). \quad (6)$$

Since $\log \binom{T_g^{i,j}}{e_g^i}$ is constant value, our final loss function $L_{\text{STRank}}^{\text{pair}}$ is as follows:

$$L_{\text{STRank}}^{\text{pair}}(x^i, x^j, e^i, e^j) = -\sum_{g=1}^{N^g} \left( e_g^i \log \hat{p}_g^i + e_g^j \log \hat{p}_g^j \right). \quad (7)$$

To construct sample pairs, we randomly select a sample from within the same tissue for each reference sample. Patch pairs are generated per tissue using grouped permutation, and the total loss for a mini-batch $M$, randomly sampled reference without considering patients, is defined to integrate relative signals across tissues as follows:

$$L_{\text{STRank}}^{\text{pair}}(M) = \frac{1}{N^b} \sum_{s=1}^{N^b} L_{\text{STRank}}^{\text{pair}}(x^i, x^{\pi(i)}, e^i, e^{\pi(i)}), \quad (8)$$

where $\pi$ denotes a permutation index obtained by randomly shuffling the sequential sample indices within each tissue $n$ and $x^{\pi(i)}$ corresponds to a randomly selected sample from the same tissue as $x^i$. After training, the $\hat{r}$ serves as a rank score indicating the relative expression levels across individual spots.

**Relation with Ranking Loss Function.** Our proposed pairwise loss function can be interpreted as a relaxed variant of a traditional ranking loss, enabling flexible optimization by considering count value while preserving the core objective of learning relative sample orderings. The previous ranking loss

functions focus solely on the relative ordering between pairs of samples, determining whether one is larger than the other. This implicitly assumes that the difference in rank scores is sufficiently large to make the ordering unambiguous. Under this assumption, $\hat{r}^i$ is sufficiently larger than $\hat{r}^j$, $\hat{r}^i \gg \hat{r}^j$ and the pairwise probabilities are treated as follows: $p_g^i = \frac{\exp(\hat{r}_g^i)}{exp(\hat{r}_g^i)+exp(\hat{r}_g^j)} \approx 1, p_g^j = \frac{\exp(\hat{r}_g^j)}{exp(\hat{r}_g^i)+exp(\hat{r}_g^j)} \approx \frac{\exp(\hat{r}_g^j)}{exp(\hat{r}_g^i)}$. Then, our loss function can be transformed as follows:

$$L_{\text{STRank}}^{\text{Pair}}(x^i, x^j, e^i, e^j) = -\sum_{g=1}^{N^g} \left( e_g^i \log \frac{\exp(\hat{r}_g^j)}{exp(\hat{r}_g^i)} \right) = -\sum_{g=1}^{N^g} \left( e_g^i \left( \hat{r}_g^j - \hat{r}_g^i \right) \right) \propto \hat{r}_g^j - \hat{r}_g^i. \quad (9)$$

Introducing a margin term and a max operation to the difference in rank scores recovers the form of conventional ranking loss functions, such as the hinge-based pairwise ranking loss.

### 3.2 Listwise STRank Loss

Similar to the pairwise approach, listwise estimation over multiple $N^k$ samples can be formulated by modeling the expression level associated with each sample. This allows the model to handle group-wise comparisons within a unified probabilistic framework.

We assume that the relationship between the list of patch images, $\mathbf{X}^{(n)} = [x^1, ..., x^{N^k}]$, extracted from the same tissue and their associated gene expression values, $\mathbf{E}^{(n)} = [e^1, ..., e^{N^k}]$, follows a multinomial distribution. This probabilistic formulation enables modeling the joint contribution of individual patches to the overall expression profile in a listwise manner.

$$\Pr(\mathbf{E}^{(n)}|\mathbf{X}^{(n)}, \mathbf{T}^{(n)}) = \prod_{g=1}^{N^g} \Pr(\mathbf{E}_g^{(n)}|\mathbf{X}^{(n)}, T_g^{(n)}), \Pr(\mathbf{E}_g^{(n)}|\mathbf{X}^{(n)}, T_g^{(n)}) = \text{Multinomial}(T_g^{(n)}, p_g^i),$$
$$(10)$$

where $T_g^{(n)} = \sum_{i=1}^{N^k} e_g^i$ is the total number of gene expression count, and $\mathbf{E}_g^{(n)} = [e_g^1, ..., e_g^{N^k}]$. The probabilities for the multinomial distribution $p_g^i$ are obtained using the softmax operation, similar to the pairwise case: $p_g^i = \frac{\exp(\hat{r}_g^i)}{\sum_{j=1}^{N^k} \exp(\hat{r}_g^j)}$.

The negative log likelihood of the multinomial distribution is transformed

$$-\log \Pr(\mathbf{E}^{(n)}|\mathbf{X}^{(n)}, T) = -\log \left( \prod_{g=1}^{N^g} \Pr(\mathbf{E}_g^{(n)}|\mathbf{X}^{(n)}, T_g) \right) \quad (11)$$

$$= -\sum_g^{N^g} \sum_i^{N^k} \left( e_g^i \log p_g^i + \log \frac{T_g!}{e_g^1! e_g^2! \cdots e_g^{N^b}!} \right). \quad (12)$$

Since the second term is independent of the model parameters, it can be omitted during optimization. The resulting listwise rank loss for spatial transcriptomics, referred to as ListWiseSTRank, is defined as follows:

$$L_{\text{STRank}}^{\text{List}}(\mathbf{X}^{(n)}, \mathbf{E}^{(n)}) = -\sum_g^{N^g} \sum_i^{N^k} e_g^i \log p_g^i. \quad (13)$$

Similarly to the pairwise loss, we define the total loss for a mini-batch $M$, which is randomly sampled without considering patients, as follows:

$$L_{\text{STRank}}^{\text{List}}(M) = \sum_{n=1}^{N^{(\text{tissuse})}} L_{\text{STRank}}^{\text{List}}(\mathbf{X}^{(n)}(M), \mathbf{E}^{(n)}(M)) \quad (14)$$

where $X^{(n)}(M)$ and $E^{(n)}(M)$ are the lists of $x$ and $e$ derived from tissue $n$ in mini-batch $M$.

**Correction Using Expression for Each Spot.** Gene expression levels can vary in detectability across spatial spots, so the total count per spot often normalizes expression data. However, such normalization converts the inherently discrete count data into continuous values, which may compromise loss

functions that rely on count-based statistical properties. To mitigate this, we introduce a correction based on the total expression level $l^i$ at each spot, enabling the model to account for inter-spot variability while preserving the count data structure. $p_g^i = \frac{\exp(\hat{r}_g^i) l^i}{\sum_{j=1}^{N^k} \exp(\hat{r}_g^j) l^j}$, where $l^i = \sum_g e_g^i$.

# 4  Experiments

We evaluated our methods using two types of datasets: synthetic datasets to confirm the hypothesis and characteristics of the proposed loss function, and real datasets to confirm practicality.

**Comparisons.** We compared our loss function with five loss functions: 1) Mean Squared Error loss (**MSE**) measures the squared difference between the predicted value and the ground truth on a per-sample basis (widely used on gene expression estimation), 2) Poisson loss (**Poisson**) models the output as a Poisson-distributed count and minimizes the corresponding negative log-likelihood for each sample, 3) Negative Binomial loss (**NB**) extends Poisson loss by incorporating a dispersion parameter to handle overdispersed count data at the individual sample level, 4) Rank loss (**Rank**) operates on pairs of samples and penalizes incorrect relative ordering, encouraging proper ranking, 5) Pearson Correlation Coefficient loss (**PCC**) is a listwise loss that maximizes the linear correlation between predicted and true values across the full batch. To examine the effectiveness of our proposed learning relative expression trends strategies, we compare the loss functions of **PairSTRank** (Section 3.1) and **ListSTRank** (Section 3.2), which are based on pairwise and listwise learning, respectively.

The Spearman Correlation Coefficient (SCC) was used as a metric to evaluate the performance.

## 4.1  Hypothesis Analysis on Synthetic Dataset.

We simulated 1D synthetic data to evaluate the effects of batch effects and stochastic noise. The reason for using synthetic data is that it is difficult to obtain ground truth from raw gene expression datasets since the observed data already contains bias and noise.

In these experiments, each input variable $x^i$ was defined as a one-dimensional scalar constrained to the interval $[0, 1]$. The corresponding gene expression level $e^i$ was modeled using a negative binomial distribution, consistent with prior work in transcriptomic data analysis [11]. Specifically, $e^i$ is also 1D data and was sampled from the distribution $\text{NB}\left(\alpha\mu(x^i) + \beta, r\right)$, where $\mu(x^i)$ is the mean response function of the input, $r$ is the dispersion parameter, $\alpha$ is scaling parameter, $\beta$ is bias parameter.

The objective of the experiments is to accurately estimate the mean function $\mu(x^i)$ from a given dataset $D = \{\mathcal{X}^{(n)}, \mathcal{E}^{(n)}\}, \mathcal{X}^{(n)} = \{x^i\}_{i=1}^{N^n}, \mathcal{E}^{(n)} = \{e^i\}_{i=1}^{N^n}$. It corresponds to finding the meaningful signal from observed data. The $\mu(x^i)$ is a nonlinear function: $\mu(x^i) = a\sin(cx^i) + b\sin(dx^i) + a + b$, which is the same with [21]. We prepared four types of functions with different parameters (*cf.* Supplementary material A). The mean SCC for the four types of functions is calculated.

We assumed a training dataset comprising two patients ($n = 2$), and each patient's gene expression is affected by experimental batch effects, which stem from sources such as differences in imaging protocols or acquisition equipment. We assumed that gene expression measurements for each patient are affected by distinct batch parameters $\alpha$ (scaling) and $\beta$ (offset). For simplicity, we set $\alpha = 1$ and $\beta = 0$ for the first patient, $\alpha = 10$ and $\beta = 10$, for the second patient (Other conditions are shown in supplementary material A). For training, 50,000 samples were independently sampled from each patient. The validation and test sets, each consisting of 10,000 samples, are sampled from a uniform distribution over the interval [0,1] in both setups. We compared a uniform situation, where input samples are sampled from a uniform distribution (Figure 2 (a)), and an imbalanced situation, where the sample of tissue 2 is sampled in the specific section (Figure 2 (b)). Figure 2 shows an example of synthetic data. The color indicates tissue, the dotted line indicates the mean function, and each plot represents an observed value $e^i$, which is sampled with the negative binomial distribution.

A simple MLP (multi-layer perceptron) with 3 linear layers ($[1 \times 128], [128 \times 128], [128 \times 1]$) with ReLU was used for the model. The epoch was 2000 using AdamW [13] with a learning rate $1e - 3$ with mini batch size = 256. For the scheduler, we used CosineAnnealing [12]. Details regarding computational resources and related settings are provided in the supplementary material B.

Table 1 shows the comparative performance of various loss functions used for gene expression estimation. Traditional pointwise losses, including widely used Mean Squared Error (MSE), optimize

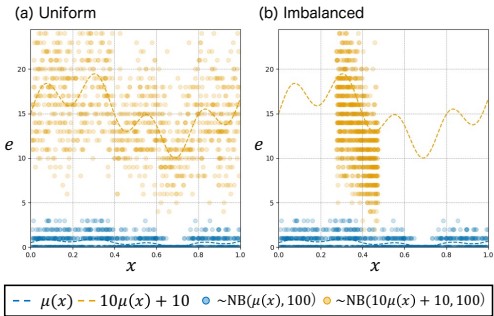

(a) Uniform    (b) Imbalanced

$-- \ \mu(x)$  $-- \ 10\mu(x)+10$  ○ $\sim NB(\mu(x),100)$  ○ $\sim NB(10\mu(x)+10,100)$

Figure 2: Example of synthetic data for validating the batch effect. Colors indicate patients; the dashed line represents the mean function to be learned; and the dots show observations. (a) Uniform setting: Each patient's data is drawn from a uniform distribution. (b) Imbalanced setting: Observed data is skewed.

Table 1: Performance comparison across multiple patient conditions. Results are based on synthetic data. Bold faces indicate the best performance in each setting, while underlined values denote the second-best.

| | | Uniform | Imbalanced |
|---|---|---|---|
| Point | MSE | 0.748 | 0.583 |
| | Po | 0.777 | 0.603 |
| | NB | 0.788 | 0.601 |
| Pair | Rank | 0.835 | 0.738 |
| | PairSTRank | 0.907 | 0.818 |
| List | PCC | 0.858 | 0.560 |
| | ListSTRank | **0.945** | **0.828** |

prediction accuracy by minimizing the absolute difference in expression levels across individual samples. In contrast, loss functions based on pairwise and listwise learning paradigms, which capture relative expression relationships between genes or samples, demonstrate superior performance overall. These results demonstrate that learning relative expression is effective in situations where batch effects are present. Notably, the proposed methods—PairSTRank and ListSTRank—consistently outperform both conventional pairwise and listwise approaches, indicating their enhanced capacity to model the structured dependencies inherent in gene expression data in both uniform and imbalanced situations. Empirical results show that ListSTRank outperforms its pairwise counterpart. We attribute this improvement to ListSTRank's ability to capture global expression patterns across entire batches, as opposed to the localized comparisons used in pairwise learning. This suggests that leveraging broader relational context is advantageous under batch-affected conditions.

## 4.2 Evaluation on Real Datasets

**Dataset.** To evaluate the effectiveness of our proposed method, we performed experiments using seven datasets from the benchmark of the HEST-1k dataset [8]: IDC, PRAD, PAAD, COAD, READ, ccRCC, and IDC-LymphNode. SKCM and LUAD datasets were excluded from the analysis because they contain only two patients and do not align with the assumptions of our study. Each dataset contains samples from three individual patients. The IDC, PAAD, and COAD datasets were acquired using the Xenium platform, whereas other datasets were obtained via the Visium platform. To avoid train/test patient-level data leakage, we used patient-stratified splits and one patient for validation and testing data, respectively, and the other patients were used for training data. The motivation for the experiment is not to compare models, but to compare loss functions. Therefore, we do not use PCC or regularization, and simply train a regression model with each loss function.

We used 50 genes with highly variable genes. To assess the influence of the loss function, we kept the feature extractor fixed and trained only a single fully connected (fc) layer, as shown in Figure 3. The feature extractor was CONCH [14], which is a vision and language foundation model for pathology. Model optimization employed the AdamW optimizer with a learning rate of $5e-5$ and a batch size of 256. We trained the model for up to 1000 epochs, with early stopping implemented using a patience threshold of 30 epochs.

Table 2 summarizes the performance comparison across all datasets. Overall, the proposed method outperforms conventional loss functions on average. Although STRank demonstrated superior performance on synthetic data, STRank did not consistently outperform alternatives across all conditions in real datasets. Because real data evaluations are based on observations that inherently include stochastic and measurement noise, it remains essential to assess whether the evaluations reliably reflect true model performance. Even under such conditions, STRank remained relatively stable and was able to demonstrate superior performance on average.

Table 2: Real dataset which is obtained from HEST-1k [8]. Bold faces indicate the best performance in each setting, while underlined values denote the second-best. Ave. is average performance.

| | Loss | IDC | PRAD | PAAD | COAD | READ | ccRCC | IDC-L | Ave. |
|---|---|---|---|---|---|---|---|---|---|
| Point | MSE | 0.393 | 0.484 | 0.307 | 0.556 | 0.140 | 0.093 | 0.168 | 0.306 |
| | Po | 0.314 | 0.485 | 0.336 | 0.524 | **0.172** | 0.091 | 0.134 | 0.293 |
| | NB | 0.199 | **0.491** | 0.119 | 0.538 | 0.160 | 0.075 | 0.126 | 0.244 |
| Pair | Rank | 0.317 | 0.317 | 0.181 | 0.566 | 0.047 | 0.059 | 0.110 | 0.228 |
| | PairSTRank | 0.494 | 0.458 | **0.346** | 0.613 | 0.136 | **0.127** | 0.228 | 0.343 |
| List | PCC | 0.472 | 0.459 | 0.307 | **0.640** | 0.105 | 0.102 | 0.198 | 0.326 |
| | ListSTRank | **0.510** | 0.459 | 0.343 | 0.597 | 0.140 | 0.125 | **0.238** | **0.345** |

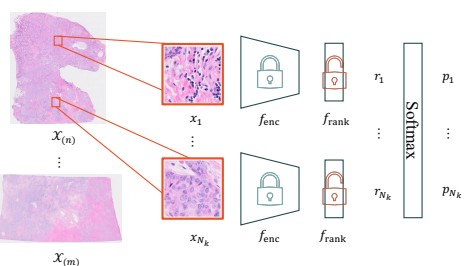

Figure 3: Illustration of our framework for the real dataset. To assess the loss function, we only update the classifier head for this evaluation.

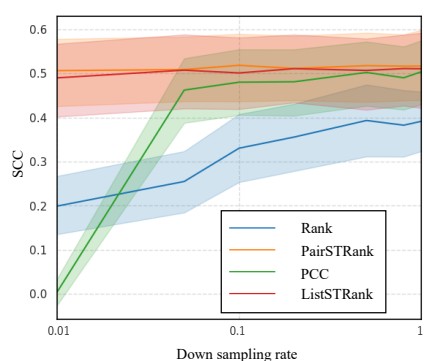

Figure 4: Effectiveness for sparsity. The x-axis is on a log scale.

## 4.3 Effectiveness for Sparcity

To further assess the robustness of STRank for the sparsity, we conducted performance variability for varying sparsities in a modified real dataset. One way to assess robustness for the sparsity is by assessing the performance on low-expressed genes. However, as discussed in [17], sparsity evaluation becomes challenging in low-expression data due to the lack of known ground-truth signals. To address this, we simulated the sparsity-enhanced expression data for genes with the top 50 highest mean expression levels. For a count of each gene in each cell, we conducted binomial sampling using a specified down-sampling rate to acquire down-sampled count data. We varied levels of expression by downsampling each gene expression count using probabilities $p = 0.01, 0.05, 0.1, 0.2, 0.5, 0.8, 1$.

Figure 4 shows the performance of the pairwise and listwise loss functions on each downsampling rate. Our Pair and List STRank loss outperforms Rank and PCC on each down-sampling rate. The difference becomes significant when the rate is $0.01$, effectiveness on highly sparse and weak signals. Since gene expression is inherently sparse, these results suggest that the STRank is well-suited for capturing gene expression signals.

## 4.4 Effect for Parameter $N^k$ of Our Loss Function

We varied $N^k$ and assessed the effect of $N^k$, which is the number of samples to calculate our ListSTRank. A larger $N^k$ is expected to be generally preferable, as it facilitates the capture of global trends. However, in the presence of noise or distortion in the global structure, pairwise learning may offer improved performance.

Table 3 shows the performance on each $N^k$. The results indicate that increasing $k$ beyond 4 leads to improved performance. However, in practice, increasing $N^k$ is not always beneficial due to numerical instability and computational overhead in both situations. For $k \geq 4$, the model demonstrates robustness, with stable performance observed at $k = 8$ and $k = 16$.

Table 3: Performance of STRank for $N^k$ on three conditions.

| $N^k$ | Uniform | Imbalanced | B Xenium |
|---|---|---|---|
| 2 | 0.907 | 0.818 | 0.447 |
| 4 | 0.938 | 0.837 | 0.462 |
| 8 | **0.958** | 0.839 | 0.455 |
| 16 | 0.943 | 0.833 | 0.458 |
| 32 | 0.938 | 0.818 | **0.463** |
| 64 | 0.926 | 0.827 | 0.462 |
| 128 | 0.941 | **0.845** | 0.457 |
| 256 | 0.945 | 0.828 | 0.459 |

## 5  Related work

**Gene expression estimation from pathological image.** Estimating gene expression from pathology images has the potential to reduce sequencing costs and help understand diseases. Deep learning has been introduced in this field, and the deep learning models are trained with patch and gene expression pairs captured by spatial transcriptomics. ST-Net [7] has introduced a transfer learning approach and estimates gene expression by a convolutional neural network pre-trained on ImageNet [5]. To utilize global information of patches, graph convolution neural networks and transformers have been introduced in Hist2gene [19] and Hist2st [30]. To effectively combine local and global information, M2OST [24] and TRIPLEX [4] effectively combined multiple features that are extracted from multiple resolutions. By focusing on the difficulty of directly estimating multiple-dimensional gene expression, exemplar-guided estimation [26, 28, 29], which utilizes retrieved gene expression, has been proposed. BLEEP [26] has trained a model with image and gene expression in a contrastive learning manner and retrieves gene expression based on the image. EGN [29] has refined the retrieved gene expression with a transformer block.

MSE loss has been mainly used as the loss function for these methods. In contrast, we focus on the relative relation among tissues and propose a novel noise-robust loss function with pairwise learning.

**Learning to rank.** Learning to rank is the field that learns the ranking function from ground-truth rankings [3]. Sculley has adapted the Stochastic Gradient Descent method for learning to rank [22], allowing models to be trained on large datasets. The ranking loss has been integrated with neural networks, which have been widely utilized in applications including Image Quality Assessment [9] and crowd counting [27, 10].

In contrast to these works, which only consider ranking, our loss function considers the size of the count value and adaptively weights depending on the relation of input samples. Since the gene expression is sparse and has a low signal, taking into account the count value helps to learn the relation in the low signal situation.

## 6  Conclusion

In this paper, we tackled gene expression estimation from pathology images by reconsidering the objective. In contrast to the previous method, which estimates the absolute expression value, we aim to learn the relative gene expression relation. In addition, we propose a novel loss function (STRank) designed to capture the relative gene expression across spatial patches by modeling the relative expression relation. Through comprehensive experiments on both synthetic and real-world datasets, we demonstrated that our method achieves more stable and reliable performance compared to traditional point-wise approaches. These results suggest that exploiting relative gene expression patterns is a promising strategy for enhancing robustness in gene expression prediction.

## 7  Limitations

We hypothesized that STRank would perform well under sparse conditions, and we evaluated its performance by varying the number of target genes from 50 to all possible genes. As shown in Supplementary D, we could not confirm the effectiveness of STRank in a sparse situation of a real dataset. Furthermore, as shown in Table 2, rigorous evaluation on real data remains an open challenge,

and the fundamental reasons why STRank fails to outperform conventional loss functions in some datasets have yet to be elucidated.

Although STRank shows effectiveness across multiple patients, it converges more slowly than PCC when evaluated on data from a single patient. This behavior is likely due to STRank's conservative weight updates under low-sample conditions. As a result, PCC may offer better convergence performance on simpler datasets with limited observations.

Another challenge is performance in multi-cohort settings, where the sample is obtained from different experimental conditions (e.g., different hospitals and procedures). As detailed in Supplementary D, our proposed loss functions exhibit limitations when applied to such multi-cohort datasets with a large number of genes. In such cases, batch effects in gene expression are amplified, and pathology images are likewise influenced by experimental batch effects due to variations in imaging conditions. While addressing this issue is beyond the scope of our study, it represents an open question for this field. Future research should aim to develop more robust and generalizable approaches to mitigate these effects.

## Acknowledgments and Disclosure of Funding

Funding in direct support of this work: JSPS KAKEN JP24KJ2205, JPMJBS2406, JP23K18509, and JP25K22846, Development (AMED) grant 24ama221609h0001(P-PROMOTE) (to YK), National Cancer Center Research and Development Fund 2024-A-6 (to YK). We used ABCI 3.0 provided by AIST and AIST Solutions.

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

# A   Details of Experiments on Synthetic Dataset

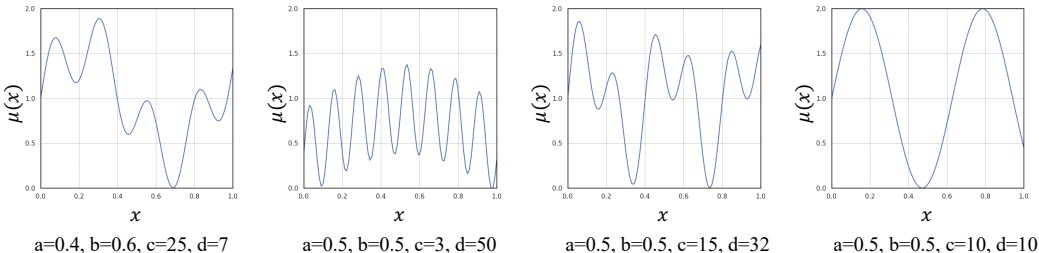

Figure 5: Visualization of $\mu(x)$.

Figure 5 shows four types of mean functions for our synthetic data. A nonlinear function was chosen to generate waveforms characterized by varying frequencies and slope gradients. This property allows the function to model complex, non-uniform signal behavior, which is relevant in representing heterogeneous patterns observed in the gene expression data.

Figure 6 shows the performance of each loss function under various parameters in the synthetic dataset. We changed scale $\alpha$, bias $\beta$, dispersion parameter $r$, scale for tissue 2 $\alpha$, bias for tissue 2 $\beta$. Overall, our loss function outperforms all comparisons on each condition. The proposed loss function demonstrates robustness under low-scale conditions. Furthermore, its effectiveness improves as the variability in intensity scales across patients increases.

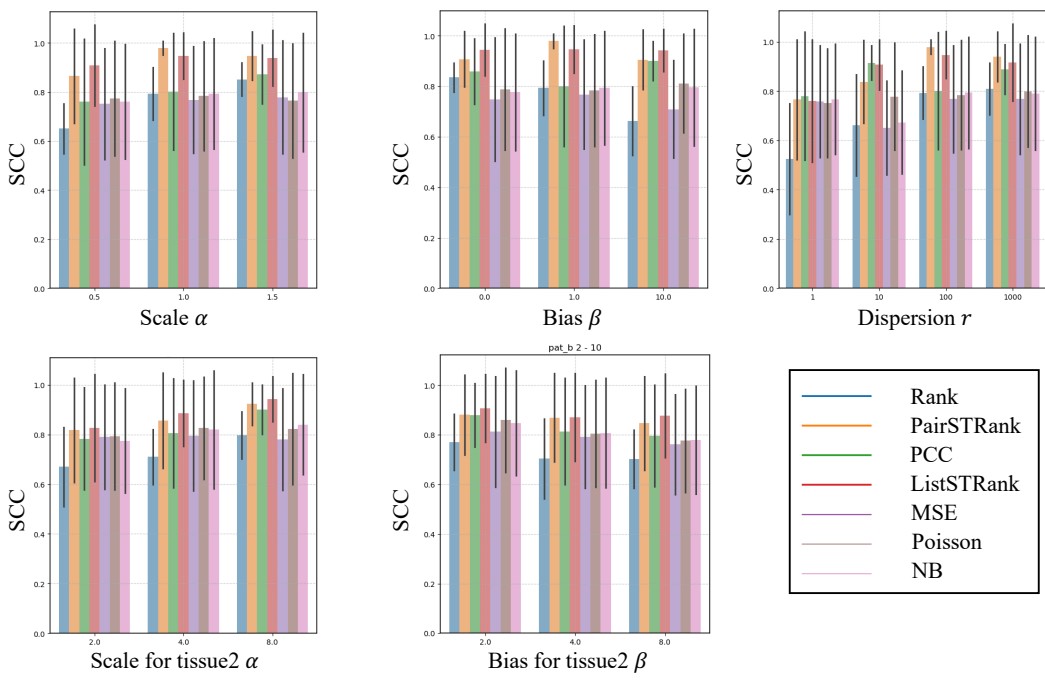

Figure 6: Visualization of $\mu(x)$.

# B   Computer Resources

We used the Cloud Environment [23] for the experiment on synthetic data, and an internal desktop computer for Experiment 2.

Experiment 1 (Cloud environment)

- CPU: 16 assigned physical CPU cores

Table 4: Performance on HER2ST dataset with 50–9385 gene sets.

| Method | 50 | 250 | 1000 | 5000 | 9385 |
|---|---|---|---|---|---|
| MSE | 0.193 | 0.181 | 0.172 | 0.162 | 0.132 |
| NB | 0.020 | 0.154 | 0.098 | 0.083 | 0.074 |
| Po | 0.009 | 0.150 | 0.095 | 0.076 | 0.069 |
| Rank | 0.095 | 0.052 | 0.041 | 0.042 | 0.018 |
| PairSTRank | 0.244 | 0.194 | 0.176 | 0.177 | 0.173 |
| PCC | 0.189 | 0.173 | 0.165 | 0.171 | 0.152 |
| ListSTRank | 0.260 | 0.175 | 0.110 | 0.085 | 0.087 |

Table 5: Performance on COAD Visium dataset with 50–2000 gene sets.

| Method | 50 | 250 | 1000 | 2000 |
|---|---|---|---|---|
| MSE | 0.2712 | 0.2090 | 0.2125 | 0.2084 |
| NB | 0.3043 | 0.1682 | 0.0899 | 0.0868 |
| Po | 0.2572 | 0.1440 | 0.0858 | 0.0829 |
| Rank | 0.1118 | 0.0714 | 0.0260 | 0.0599 |
| PairSTRank | 0.3456 | 0.1957 | 0.1383 | 0.1319 |
| PCC | 0.2615 | 0.2000 | 0.2035 | 0.1951 |
| ListSTRank | 0.3398 | 0.1973 | 0.1404 | 0.1375 |

- GPU: None
- Memory: 320 GB

Experiment 2 (Internal desktop environment)

- CPU: 12th Gen Intel(R) Core(TM) i9-12900KS, Physical Cores: 16
- GPU: NVIDIA RTX A6000
- Memory: 128 GB

## C  Licenses for Existing Assets

We implemented our method with Pytorch [20] with modified BSD LICENSE, PytorchLightning [6] with Apache-2.0 LICENSE. For the feature extraction from whole slide images, we modified the CLAM implementation [15]. We used Hest 1k [8] with CC BY-NC-SA 4.0 for the real datasets.

## D  Experiments on More Realistic Scenarios

We conducted experiments using the HER2ST datasets [1] and the COAD Visium (multi-cohort) with large gene sets to evaluate the robustness of our loss function on more realistic gene expression estimation scenarios. For HER2ST, we used gene sets of size 50, 250, 1000, 5000, and the full set of 9,385 genes. For COAD, we evaluated the loss functions on gene sets of size 50, 250, 1000, and 2000.

The results are shown in Tables 4 and 5. Our loss function demonstrated robustness on the HER2ST dataset, which is a highly sparse spatial transcriptomics dataset, outperforming other loss functions. This suggests that the proposed loss function is effective for sparse real-world data and for mitigating batch effects within the same cohort. However, for the COAD dataset, our loss function underperformed MSE and PCC when using multi-cohort data with large gene sets (250–2000 genes).

In multi-cohort settings (*e.g.,* COAD setting in Table 5) with many target genes, we assume that the distribution of stochastic noise varies substantially across cohorts; therefore, for low-signal genes where the noise component is dominant, further extensions that account for cohort-specific noise characteristics are needed. To effectively handle multi-cohort data, additional factors beyond batch

Table 6: Results of different methods on Hest2gene

| Method | Hest2gene |
|--------|-----------|
| MSE | 0.039 |
| NB | 0.005 |
| Po | -0.014 |
| Rank | 0.042 |
| PairSTrank | 0.052 |
| PCC | 0.039 |
| ListSTrank | 0.046 |

effects should be considered, including potential differences in biological signals and domain shifts in image features. This is one of the open problems and is essential for the practical application of gene expression prediction.

## E  Integration with Previous Method

To evaluate the generalizability of the proposed loss function, we examined its performance when integrated into existing gene expression estimation methods. We evaluated the Spearman correlation coefficient (SCC) for predicting 250 genes using the HER2ST dataset [1]. In contrast to the original study, we adopted a patient-level data split to evaluate generalizability. We then assessed the performance of HisToGene [19] trained with various loss functions. As summarized in the Table 6, the proposed PairSTrank loss achieved the highest performance. In conclusion, our loss functions demonstrated effectiveness on this setup.

## F  Exploring the Impact of Mini-Batch Sampling Strategies

To investigate the effect of a mini-batch sampling strategy, we compared three settings: the original implementation (Default), a setting where the mini-batch contains samples from only one tissue (Intra-tissue), and a setting where samples are evenly sampled from all tissues (Inter-tissue).

Table 7 shows the results of these experiments on the 250 and 1000 gene sets. Even when we use different mini-batch strategies, the performance of ListSTRank is not improved. PairSTRank loss also does not consider the intra-tissue and inter-tissue, but it still outperforms ListSTRank loss. This suggests that the performance degradation of ListSTRank is not due to the mini-batch setting. Based on these results, we suspect that the performance degradation of ListSTRank may be due to numerical effects.

Table 7: Effect of mini-batch sampling on 250 and 1000. We compare three settings: Default, Intra-tissue (mini-batch contains samples from only one tissue), and Inter-tissue (samples are evenly sampled from all tissues).

| Method | 250 | 1000 |
|--------|-----|------|
| Default | 0.175 | 0.110 |
| Intra-tissue | 0.177 | 0.105 |
| Inter-tissue | 0.173 | 0.105 |

