# OpenReview forum: "Learning Relative Gene Expression Trends from Pathology Images in Spatial Transcriptomics"
_NeurIPS.cc/2025/Conference — NeurIPS 2025 poster_

### Official Review · Reviewer_XD8s · 2025-07-03

**Clarity:** 3
**Significance:** 4
**Originality:** 3
**Rating:** 5
**Confidence:** 4

**Summary:**

This paper introduces a new loss/objective function to train regression models for Spatial gene-expression estimation from histology images. While the application of the objective function is on spatial transcriptomics data, the objective function seems to be quite generalizable for any kind of data with stochastic noise and/or batch effects. The authors suggest, due to the fairly assumed nature of spatial gene expressions where the relative values of the genes between two biologically different regions (eg. Cancerous & non-cancerous), exhibit a pattern across multiple experiments, this pattern can be exploited to make the gene-expression prediction models more robust to such batch effects and stochastic noise. The paper elaborates on certain decisions of the loss function, and shows performance of the objective function using synthetic data experiments, real world spatial xenium/visium data experiments and further ablation studies. The paper also shows effectiveness of the loss function in relation to sparsity of the data. The motivation of the paper is to compare the loss function to other established loss functions and their effectiveness on the downstream task and not to compare different models.

**Questions:**

Scalable comparisons: Can the authors compare the performance of the STRank loss function against other losses for a larger dataset? This will make the paper even stronger. For example, the main purpose of HEST to use a small set of samples was to compare different histology foundation models, irrespective of the inherent noise in the spatial dataset. Since your work is to compare loss functions with the main motivation to train models robust to batch effects, it will be highly valuable to see the effectiveness of the model on a larger distribution of batch effects and stochastic noise. This can be achieved by including more samples in the mentioned 7 tissue tasks from the HEST dataset.

Number of genes > 50 in real world data (Can be combined with previous question):  Can the authors show the effectiveness of the loss function on a larger set of genes? As suggested, the experiment can be scaled to ~1000 HVG genes for Visium and the complete gene-panel for Xenium which ranges between 280 - 480 genes in the HEST dataset.

Application of loss function on other possible architectures: Since the loss function can be used instead of conventional regression based losses like MSE, can the authors show the results of the loss function when applied to other feasible methods like ST-net to see whether it improves the gene-expression prediction performance of the method?

Synthetic experiment val and test splitting confusion: In the synthetic data setting, the validation and test set is sampled from the same distribution as the training set for both the Uniform and Imbalanced setting. For the imbalanced setting, it’s confusing whether the validation and test sets were sampled from the same section of Tissue 2 or a different section. If it’s sampled from the same section of Tissue 2, one could say the model has seen the test samples. Please clarify about the same.

Results for the above questions can highly boost the impact of the paper.

**Ethical Concerns:**

["NO or VERY MINOR ethics concerns only"]

**Final Justification:**

The authors addressed my concerns in a strong rebuttal, I especially appreciate the additional results.

**Limitations:**

The limitations were addressed based on the real-world data experiments performed which aren’t rigorous. The overall suggestion for improving this paper is to show results for larger datasets and comparisons with other models.

**Paper Formatting Concerns:**

Few equation variables like N^b have not been explained and had to be understood from context.

**Quality:**

2

**Strengths And Weaknesses:**

The paper is well written and focuses on the problem statement clearly. It addresses a well known problem of batch-ridden data and devises a well thought and justified solution for the same. The authors clearly explain certain assumptions and decisions taken when developing the loss function. Most of the experiments performed indicate the benefits gained from the suggested solution i.e new loss function.The approach of modelling pair-wise or list-wise gene expression inputs in a probabilistic distribution and estimating the relative rank of the genes based on this objective is novel, and the synthetic dataset experiments adds value to show the effectiveness of using the loss during training. The overall effectiveness of the loss function is modest and well put.

While the synthetic experiments are well thought and show clear effectiveness of the approach, the real world data experiments are limited. The authors show the gene expression prediction experiment on merely 50 HVGs similar to the HEST paper, which is not enough to show the difference in performance of different loss functions. As mentioned in related works, several methods for gene-expression prediction have been developed over time (Hist2Gene, BLEEP, Hist2st) showing efficacy of their approaches on a much larger set of genes. The authors miss an opportunity to test their loss function on a larger set of genes where MSE can be lacking in performance. This experiment can easily be scaled to ~1000 HVG genes for Visium (with sparsity in high-expressed genes) and the complete gene-panel for Xenium which ranges between 280 - 480 genes in the HEST benchmark dataset. It would be curious to see the effectiveness of the STRank loss function in this range of genes. Additionally, the experiment of sparsity is ambiguous in showing the effectiveness of the approach. The experiments in the main text show the loss function performs robustly in case of sparse data, but as mentioned in the supplements and limitations, the hypothesis doesn’t hold true for a larger set of genes which negates the claims in the results section 4.3.

The limitations of the loss function are addressed, based on the experiments performed. But the concern about the true effectiveness of the loss function still remains when scaled up to larger datasets which is possible by utilising other samples in the HEST dataset.

---

> ### Author Rebuttal · Authors · 2025-07-30
>
> We thank your careful review and insightful comments. We appreciate your recognition of the clarity of our problem formulation, the justification of our proposed loss function, and the novelty of modeling gene expression in a probabilistic ranking framework. We address reviewer comments below and will incorporate all feedback.
> ### Q1: Can the authors compare the performance of the STRank loss function against other losses for a larger dataset?
> To demonstrate the scalability of the proposed method, we conducted experiments on the HER2ST [1] dataset, which contained 6 patients with 36 whole slide images. The number of target genes for prediction was varied across three settings: 50, 250, and 1000.
>
> Table 1 shows the result of the comparisons. Among all evaluated loss functions, PairSTRank consistently achieved the best performance on average. While ListSTRank demonstrated good performance on synthetic data and the 50-gene set condition, the performance of ListSTRank deteriorated as the number of target genes increased.
>
> ListSTRank exhibited best performance across both the synthetic dataset and the experiments described in Section 4.3, suggesting robustness to noise and batch effects. In addition, PairSTRank maintained consistently high performance even as the number of target genes increased. These observations support the hypothesis that the observed degradation in ListSTRank's performance stems from numerical instability, rather than from noise or batch-related artifacts.
> This issue lies beyond the scope of this study, which addresses stochastic noise and batch effect. We will include these results and the discussion in the supplementary material of the camera-ready version.
>
> Table 1: Performance across different numbers of target genes (50, 250, 1000) on HER2ST dataset [1].
> | Method       | 50  | 250 | 1000 |
> |--------------|--------|--------|--------|
> | MSE          | 0.193  | *0.181*  | *0.172*  |
> | NB           | 0.020  | 0.154  | 0.098  |
> | Po           | 0.009  | 0.150  | 0.095  |
> | Rank         | 0.095  | 0.052  | 0.041  |
> | PairSTrank   | *0.244*  | **0.194**  | **0.176**  |
> | PCC          | 0.189  | 0.173  | 0.165  |
> | ListSTrank   | **0.260**  | 0.175  | 0.110  |
>
> [1] Andersson+, Spatial deconvolution of HER2-positive breast tumors reveals novel intercellular relationships, 2020.
>
> ### The validity of our original experiments.
> As noted by you, our proposed loss function is expected to demonstrate effectiveness in settings with a large number of patient samples. However, many existing gene expression datasets, such as Her2ST [1], are based on earlier spatial transcriptomics technologies, which contain more observation noise than the Xenium platform employed in our study.
> Real-world gene expression data includes significant observational noise, and the ground truth signal is typically unknown. Reliable evaluation thus requires data in which the underlying biological signal is at least partially preserved. Therefore, we believe the dataset captured by Xenium is better than the dataset captured by Spatial Transcriptomics. Furthermore, as discussed in [2, 3], evaluation procedures such as the use of smoothing introduce additional methodological variability, raising concerns about reproducibility and trustworthiness.
>
> To mitigate these issues, we adopt the hest 1k benchmark, which is constructed primarily from Xenium-based data. This platform offers more stable and accurate measurements compared to earlier spatial transcriptomics techniques, making it more suitable for evaluating the effectiveness of our loss function.
>
> For these reasons, we consider our experimental design using real-world Xenium data to be valid and appropriate for demonstrating the strengths of our proposed method.
>
> [2] Mejia+, SpaRED benchmark: Enhancing Gene Expression Prediction from Histology Images with Spatial Transcriptomics Completion, MICCAI 2024
>
> [3] Ganguly+, MERGE: Multi-faceted Hierarchical Graph-based GNN for Gene Expression Prediction from Whole Slide Histopathology Images, CVPR 2025
>
> ### Q3: Can the authors show the results of the loss function when applied to other feasible methods?
> We evaluated the Spearman correlation coefficient (SCC) for predicting 250 genes using the HER2ST dataset. In contrast to the original study, we adopted a patient-level data split to better reflect generalization across individuals. We then assessed the performance of HisToGene[4] trained with various loss functions. (Since STnet is almost the same as our architecture, we selected HisToGene) As summarized in the table, the model incorporating the proposed PairSTrank loss achieved the highest performance. In conclusion, while we do not perform cross-validation, our loss functions demonstrated effectiveness on this setup.
> Table 2: Performance of loss function on HisTogene
> | Method        | Hest2gene   |
> |---------------|---------|
> | MSE           | 0.039   |
> | NB            | 0.005   |
> | Po            | -0.014  |
> | Rank          | 0.042   |
> | PairSTrank    | 0.052   |
> | PCC           | 0.039   |
> | ListSTrank    | 0.046   |
>
> [4] Pang+, Leveraging information in spatial transcriptomics to predict super-resolution gene expression from histology images in tumors, 2021
>
> ### Q4: Synthetic experiment, val and test splitting confusion.
> In both settings, validation and test data are sampled from a uniform distribution over the interval [0,1]. Thank you for pointing that out. As you pointed out, the term “generated from the same distribution” lacked clarity, and we will revise it in the camera-ready version.

---

> > ### Comment · Reviewer_XD8s · 2025-08-05
> >
> > I thank the authors for their thoughtful and detailed responses. While they have certainly helped clarify many aspects of the work and I am satisfied with Q3 and Q4 responses, I still have concerns regarding Q1 and Q2.
> >
> > Regarding Q1, while I appreciate the additional results on the HER2ST dataset and acknowledge the insights provided into STRank's performance, I believe the core intent of my question about robustness under diverse batch effects and stochastic noise remains incompletely addressed. The HER2ST dataset, despite its size in total samples, represents data from only 6 patients with consecutive sections - effectively a single-cohort study with limited batch heterogeneity. My primary concern was evaluating STRank under more realistic and diverse batch settings that arise when pooling multiple datasets from different acquisition sites, technologies, or patient populations - a key motivation for developing batch-robust losses. Given that HEST was specifically designed to provide this multi-cohort structure, assessing STRank's performance on organ-specific tasks within HEST using samples across cohorts (e.g., human kidney Visium samples [n=72 across 7 cohorts] or COAD Visium samples [n=73 across 7 cohorts]) would provide a more challenging and relevant test of batch robustness. Such analysis would substantially strengthen the generalizability claims.
> >
> > Regarding Q2, I appreciate the experiment demonstrating the loss function's effectiveness on larger gene sets. While I better understand the results now and acknowledge PairSTRank's ability to scale with increasing gene numbers, I find the explanation that "ListSTRank's performance degradation stems from numerical instability rather than noise or batch-related artifacts" not fully convincing. Given that you use a mini-batch size of M where each element comes from a different tissue, it seems plausible that the model struggles to learn only the relative differences within the patch list due to extreme variations between patches in different tissues. I would be particularly interested in ListSTRank's performance on larger gene sets (250, 1000) when batch size M = 1.
> >
> > Additionally, I remain unconvinced by the sparsity experiment. While I agree that Xenium generally has less noise compared to Visium, the sparsity experiment was conducted on only 50 HVGs, which in a spatial context may not fully represent biological variability. This raises questions about the loss function's performance on both a broader batch distribution and larger gene sets in combination. These questions could be addressed using either multi-cohort Visium samples with whole transcriptome (≥1000 genes) or multi-cohort Xenium samples with full gene-panel (~380-480 genes).
> >
> > I will revisit my score once these points are more thoroughly addressed.

---

> ### Author Response · Authors · 2025-08-06
> **Regarding the second question.**
>
> Thank you for your comments. I would appreciate some clarification regarding one point.
> You mentioned: “I would be particularly interested in ListSTRank’s performance on larger gene sets (250, 1000) when batch size M = 1.”
>
> Since M represents the batch size, setting M = 1 would make listwise learning infeasible. Do you assume that the performance degradation is caused by the difference in the number of pair or list samples per each tissue, and are you assuming an experiment that using one pair or list samples per each tissue?
>
> We would like to understand your concern precisely. Regarding other concerns, we will reply together after summarizing the points.

---

> > ### Comment · Reviewer_XD8s · 2025-08-06
> >
> > Let me clarify my understanding of the **listwise learning** setup to ensure we're aligned:
> >
> > For each tissue $n$, you have a list of $s$ samples $X^{(n)} = [x^{1}, \dots, x^{s}]$ with corresponding expression values $E^{(n)} = [e^{1}, \dots, e^{s}]$.
> >
> > While **pairwise learning** uses $s = 2$ samples per tissue, **listwise learning** considers $s > 2$ samples. When using batch size $M$, you process $M$ different tissues in parallel, each with their own list of $s$ samples.
> >
> > Given this setup, I wonder if the reported performance degradation might be related to how the model handles **inter‑tissue** versus **intra‑tissue** variations. Specifically, would experiments with $M = 1$, focusing on one tissue's list at a time, help determine whether the model is being disproportionately influenced by variations between tissues rather than learning the more subtle relative differences within each tissue's samples?

---

> > > ### Author Response · Authors · 2025-08-07
> > >
> > > Thank you for clarifying your question. In our implementation of the ListSTRank loss, we first sample M samples from the entire dataset regardless of tissue, and then split these into patient-specific subsets $\mathbf{X}^{(n)}(M), \mathbf{E}^{(n)}(M)$ to compute the loss. We recognize that our original description was unclear and will revise the manuscript accordingly.
> > >
> > > As you pointed out, the current implementation does not explicitly distinguish between inter-tissue and intra-tissue variations, and this distinction is left to the mini-batch sampling process. We agree that the observed performance degradation of the ListSTRank loss may be due to the mini-batch processing, as you suggested. We will further investigate this issue, including the concerns you raised, as much as time permits.

---

> ### Author Response · Authors · 2025-08-09
>
> ### Q1: A more rigorous evaluation of STRank loss for batch robustness on multi-cohort datasets like COAD Visium in HEST.
>
> We conducted experiments on the COAD Visium samples from HEST 1k, which includes samples from 7 different cohorts with 50 HVGs.
> Table 1 shows the performance of comparisons. The results indicate that PairSTRank and ListSTRank outperform all other loss functions.
>
> Table 1: Performance on COAD visium samples in HEST 1k.
> | loss_function | COAD visium |
> |---------------|-------------|
> | MSE           | 0.271       |
> | NB            | 0.304       |
> | Po            | 0.257       |
> | Rank          | 0.112       |
> | PairSTRank    | **0.346**       |
> | PCC           | 0.261       |
> | ListSTRank    | 0.340       |
>
>
>
> While most previous studies [1, 2, 3] have evaluated their methods using datasets from a single ST technology, our experiments include benchmarks on HEST 1k, which contains multiple organ sites and ST technologies: Xenium and Visium data, and the spatial transcriptomics in this rebuttal. Compared to prior work, we believe our evaluation demonstrates sufficient generalizability across different ST technologies and organ sites.
>
>
> > [1] Minxing+, Leveraging information in spatial transcriptomics to predict super-resolution gene expression from histology images in tumors, 2021
> > [2] Ganguly+,  MERGE: Multi-faceted Hierarchical Graph-based GNN for Gene Expression Prediction from Whole Slide Histopathology Images, CVPR 2025.
> > [3] Xie+, Spatially Resolved Gene Expression Prediction from H&E Histology Images via Bi-modal Contrastive Learning, Neurips 2023.
>
> As you pointed out, developing methods that are robust to different acquisition sites, technologies, or patient populations is indeed a mandatory direction for addressing truly realistic scenarios. However, when integrating data across cohorts, additional challenges arise beyond the batch effects in gene expression data, such as domain gaps in image modalities caused by differences in staining protocols. Furthermore, for integrating different ST technologies, issues such as differences in resolution and field of view introduce further complexity beyond simple batch effects. We consider these issues to be open questions for the field of gene expression estimation. Our method does not fully solve these issues, but it was designed to mitigate batch effects more effectively than existing approaches, and we believe our work provides a clear first step toward addressing realistic and diverse batch effects.
>
>
> ### Q2: Further investigation of ListSTRank loss by focusing on intra-tissue and inter-tissue.
> To investigate the impact of the number of tissue samples within a mini-batch, we compared three settings: the original implementation (Default), a setting where the mini-batch contains samples from only one tissue (Intra-tissue), and a setting where samples are evenly sampled from all tissues (Inter-tissue).
>
> Table 2 shows the results of these experiments on the 250 and 1000 gene sets. Even when we use different mini-batch settings, the performance of ListSTRank is not improved.
> PairSTRank loss also does not consider the intra-tissue and inter-tissue, but it still outperforms ListSTRank loss. This suggests that the performance degradation of ListSTRank is not due to the mini-batch setting. We will add this analysis to the supplementary material.
>
> Table 2: Effect of mini-batch sampling on 250 and 1000. We compare three settings: Default, Intra-tissue (mini-batch contains samples from only one tissue), and Inter-tissue (samples are evenly sampled from all tissues).
> | Method       | 250 | 1000 |
> |--------------|--------|--------|
> | Default   | 0.175 | 0.110 |
> | Intra-tissue   | 0.177 | 0.105 |
> | Inter‑tissue  | 0.173  | 0.105 |
>
>
>
> Due to the character limitation of comments, we reply the third question below.

---

> ### Author Response · Authors · 2025-08-09
>
> ### Q3: Validity of the sparsity experiment and experiments using large gene sets.
>
> #### Validity of our sparsity experiment.
> When the observed data is sparse, it is difficult to verify whether the model can capture the biological signal because the composition of the biological variable signal and the noise component cannot be determined. Therefore, we simulated a sparse situation in the sparsity experiment.
> The observed count contains a biological signal and observation noise, and in sparse cases, the noise component is dominant.
> To properly evaluate the ability to capture biological signals under sparse conditions, we believe it is appropriate to simulate scenarios where the original biological signal is known and then add noise to create sparsity. This approach allows us to directly assess whether the model can recover the underlying signal despite the presence of stochastic noise and batch effects. In our sparsity experiment, we simulated a scenario where the observed data is sparse due to the scale of the assumed batch effect and stochastic noise by down-sampling the signal of 50 highly variable genes (HVGs) that are expected to have variable signals.
>
> While our experiment does not comprehensively address all challenges associated with sparsity in real-world data, it demonstrates that our method is effective in scenarios where data sparsity is primarily caused by stochastic noise, which is our main motivation.
> We acknowledge that further investigation is needed to fully assess robustness under broader and more complex sources of sparsity, such as those arising from biological heterogeneity or technical artifacts in multi-cohort datasets.
>
> #### Experiments using large gene sets.
> We conducted experiments using the HER2ST datasets and the COAD Visium (multi-cohort) with large gene sets to evaluate the robustness of our loss function.
> For COAD, we evaluated the loss functions on gene sets of size 50, 250, 1000, and 2000.
> For HER2ST, we used gene sets of size 50, 250, 1000, 5000, and the full set of 9,385 genes.
>  The results are shown in Tables 3 and 4.
> Our loss function demonstrated robustness on the HER2ST dataset, which is a highly sparse spatial transcriptomics dataset, outperforming other loss functions. This suggests that the proposed loss function is effective for sparse real-world data and for mitigating batch effects within the same cohort. However, for the COAD dataset, our loss function underperformed MSE and PCC when using multi-cohort data with large gene sets (250–2000 genes). In multi-cohort settings with many target genes, we assume that the distribution of stochastic noise varies substantially across cohorts; therefore, for low-signal genes where the noise component is dominant, further extensions that account for cohort-specific noise characteristics are needed.
> To effectively handle multi-cohort data, additional factors beyond batch effects should be considered, including potential differences in biological signals and domain shifts in image features. We agree that these considerations are essential for the practical application of gene expression prediction, and we will include this discussion in the Limitations and Supplementary Materials.
>
> Table 3: Performance on HER2ST dataset with 50-9385 gene sets.
>
> | Method       | 50  | 250 | 1000 | 5000 | 9385 |
> |--------------|--------|--------|--------|--------|--------|
> | MSE          | 0.193  | 0.181  | 0.172  | 0.162 | 0.132 |
> | NB           | 0.020  | 0.154  | 0.098  | 0.083 | 0.074 |
> | Po           | 0.009  | 0.150  | 0.095  | 0.076 | 0.069  |
> | Rank         | 0.095  | 0.052  | 0.041  | 0.042 | 0.018  |
> | PairSTRank   | 0.244  | **0.194**  | **0.176**  | **0.177** | **0.173** |
> | PCC          | 0.189  | 0.173  | 0.165  | 0.171 | 0.152  |
> | ListSTRank   | **0.260**  | 0.175  | 0.110  | 0.085 | 0.087  |
>
> Table 4: Performance on COAD Visium dataset with 50-2000.
>
> | Method     | 50     | 250    | 1000   | 2000   |
> |------------|--------|--------|--------|--------|
> | MSE        | 0.2712 | **0.2090** | **0.2125** | **0.2084** |
> | NB         | 0.3043 | 0.1682 | 0.0899 | 0.0868 |
> | Po         | 0.2572 | 0.1440 | 0.0858 | 0.0829 |
> | Rank    | 0.1118 | 0.0714 | 0.0260 | 0.0599 |
> | PairSTRank  | **0.3456** | 0.1957 | 0.1383 | 0.1319 |
> | PCC   | 0.2615 | 0.2000 | 0.2035 | 0.1951 |
> | ListSTRank | 0.3398 | 0.1973 | 0.1404 | 0.1375 |
>
> While our method does not perfectly resolve every possible batch effect or noise source arising from the full diversity of realistic datasets, we focus on two critical challenges in this field: batch effects and stochastic noise. We believe that successfully addressing these issues within the same cohort represents a meaningful and substantial contribution compared to prior work. We are confident that this study advances research in this domain by tackling these fundamental obstacles.

---

### Official Review · Reviewer_NEgj · 2025-07-03

**Clarity:** 3
**Significance:** 2
**Originality:** 2
**Rating:** 3
**Confidence:** 3

**Summary:**

The manuscript introduces STRank, a pair-wise/list-wise loss designed to predict relative gene-expression ranks from pathology image patches rather than absolute counts, with the aim of being inherently robust to batch effects and stochastic noise in spatial-transcriptomics (ST) measurements. The authors motivate the method, derive binomial/multinomial likelihoods for pair-wise and list-wise settings, and benchmark STRank against point-wise (MSE, Poisson, NB), pair-wise (Rank) and list-wise (PCC) objectives on synthetic toy data and the HEST-1k benchmark. On average, STRank outperforms the baselines on synthetic data and shows modest gains on real datasets.

**Questions:**

Could you discuss the computational complexity of the proposed model and compare with SOTA models?

**Ethical Concerns:**

["NO or VERY MINOR ethics concerns only"]

**Final Justification:**

I increased score to 3 based on the additional experimental evidence.

**Limitations:**

The paper presents an elegant loss tailored to batch/noise robustness and shows promise on synthetic data. However, the experimental evidence on real ST tasks is limited, and, critically, the method omits spatial context—the defining feature of spatial transcriptomics. Incorporating (or justifying the exclusion of) spatial information and benchmarking against modern ST foundation models would greatly improve the paper.

**Paper Formatting Concerns:**

No formatting concerns.

**Quality:**

3

**Strengths And Weaknesses:**

Strengths:
- Framing expression estimation as a ranking task is intuitive given scaling biases across batches and low S/N ratios in ST counts.
- The proposed loss function, STRank can be attached to any differentiable backbone without architectural changes, making it easy to adopt

Weaknesses:
- Despite being framed as a spatial transcriptomics problem, STRank neither models spot coordinates nor uses tissue-level spatial graphs/priors. Spatial context is critical in ST. The spatial information can be used to select samples in list wise ranking.
- Baselines do not include current ST foundation models such as cGPT-spatial. Simple linear heads on frozen CONCH features are compared instead, leaving open whether STRank still helps when stronger domain-specific encoders are used.
- The authors do not evaluate by fine-tuning scGPT-spatial with STRank loss, which would be crucial in knowing if the loss actually boosts an existing ST foundation models.

---

> ### Author Rebuttal · Authors · 2025-07-29
>
> We thank the reviewer for taking the time to review our manuscript. We are encouraged by your assessment of our setup as intuitive and by your recognition of the method’s general applicability. We address your comments and concerns below.
>
>
> ### Q1: Why does the proposed method omit the use of spatial contextual information?
> The objective of our method is to propose a loss function which are robust to stochastic noise and batch effect: it is not to propose a new architecture designed to capture spatial context, as done in prior work. As discussed in [1, 2], methods that use spatial context may introduce estimation biases, which can obscure the isolated effect of the proposed loss function. To enable a clearer evaluation of our loss function, we therefore opted to use a simple gene expression estimation model. To the best of our understanding, this is a common practice when evaluating target functions.
>
> [1] Mejia+, SpaRED benchmark: Enhancing Gene Expression Prediction from Histology Images with Spatial Transcriptomics Completion, MICCAI 2024
>
> [2] Ganguly+, MERGE: Multi-faceted Hierarchical Graph-based GNN for Gene Expression Prediction from Whole Slide Histopathology Images, CVPR 2025
>
> ### Q2: Why do we not use scGPT-spatial for feature extraction?
> Thank you for your suggestion regarding scGPT-spatial. Does reference [3] correctly refer to your suggested scGPT-spatial? If [3] indeed corresponds to scGPT-spatial, the method seems to be designed for feature extraction for gene expression and is difficult to use it for image feature extraction. Therefore, we could not apply scGPT-spatial for gene expression estimation from a pathology image. However, if you indicate a different study, we would appreciate it if the formal title of the correct publication could be provided for clarification.
>
> [3] Wang+, scGPT-spatial: Continual Pretraining of Single-Cell Foundation Model for Spatial Transcriptomics
>
> ### Q3: Could you discuss the computational complexity of the proposed model and compare it with SOTA models?
> Since our proposed model consists only of a classification head, it contains fewer parameters compared to existing methods. We provide a table summarizing the number of parameters for each method. However, we emphasize that our primary contribution lies not in architectural design, but in the task formulation and the proposed loss function. Therefore, model complexity is not the central focus of our work.
>
> | Method    | Number of Parameters |
> |-----------|----------------------|
> | Ours      | 128,000              |
> | HE2GENE   | 73,281,512           |
> | TRIPLEX   | 29,294,296           |

---

### Official Review · Reviewer_o7H2 · 2025-07-03

**Clarity:** 3
**Significance:** 2
**Originality:** 3
**Rating:** 5
**Confidence:** 4

**Summary:**

The authors study the problem of predicting spatially resolved gene-expression from histopathology images.

Instead of directly predicting the absolute expression levels, they aim to predict the _relative_ expression relationships. To this end, they propose a loss function called STRank which models the gene-expression as a discrete distribution conditioned on pair-wise or list-wise inputs.

The proposed STRank is compared to five other loss functions on a synthetic 1D dataset, and on 7/9 tasks from the HEST-1k dataset.

**Questions:**

Questions/Suggestions:
- Could you please clarify what you mean by "Learning to Relative Expression" in the title and "learning to relative expression strategies"?
- As a practitioner, how do I know which one of PairSTRank or ListSTRank to use for a given problem? In what type of scenarios is e.g. PairSTRank preferable? Why?
- Given that your models don't output predictions for the absolute gene-expression levels, how should it actually be used in practice? Given a WSI for e.g. a new breast cancer patient, for whom I want to analyze the gene-expression of the PAM50 genes, how do you apply your model? What clinically relevant tasks can your models be used for? How?
- In Table 2 you only report results in terms of the Spearman correlation, while the HEST-1k benchmark uses the Pearson correlation (see e.g. https://github.com/mahmoodlab/HEST?tab=readme-ov-file#hest-benchmark-results-083024). Your results and the results for CONCH in the benchmark are quite different (and perhaps this is just because you report Spearman and they Pearson), could you report Pearson correlation results for Table 2 as well, to enable a direct comparison with the benchmark?




Minor things:
- Inconsistent use of capitalization in section headings, e.g. "2 Setup and notation" and "5 Related work" vs "3 Spatial Transcriptomics Ranking Loss", and "4.1 Hypothesis analysis on synthetic dataset" vs "3.1 Pairwise ST Rank Loss".
- Line 27 - 29: MSE is defined twice. And then again on line 93. And then yet again on line 225.
- Line 45: "While learning to rank (i.e., ranking loss) is one of the solutions to learn the relationships between samples. This ranking loss..." --> "Learning to rank (i.e., ranking loss) is one possible solution to learn the relationships between samples. This ranking loss...", perhaps?
- Line 46: "This ranking loss learns which of a given pairwise of samples has a higher" --> "This ranking loss learns which of a given pair of samples has a higher"?
- Line 80: "without consider tissue n" --> "without considering tissue n"?
- Line 115: "Similar to learning to rank setting" --> "Similar to the learning to rank setting"?
- "3.1 Pairwise ST Rank Loss" --> "3.1 Pairwise STRank Loss", "3.2 Listwise ST Rank Loss" --> "3.2 Listwise STRank Loss"?
- Line 118: "Let me consider" --> "Let us consider"?
- End of equation 8: The period should be replaced with a comma?
- Line 151: Inconsistent indices, r^{i} but r_{j}.
- "SCC" should probably be defined on line 193, before being used on Line 207 etc.
- Line 222: Not sure what "ConeAnnealing" is? Cosine?
- Line 245: "We train simple regression."?
- Throughout the paper, a bit inconsistent use of "STrank" vs "STRank".
- "HEST 1k" and "Hest 1k" should instead be "HEST-1k"?
- In Figure 3, both WSIs are marked as X_{(n)}, should the indices perhaps different?
- Line 273: "Since gene expression is sparse, our loss is suit. Since gene expression is inherently sparse, these results suggest that the STRank is well-suited for capturing gene expression signals", probably forgot to remove the first short sentence "Since gene expression is sparse, our loss is suit"?
- Line 290: "To utilize global information of patch, graph convolution neural network, and transformer have been introduced on Hist2gene [15 ] and Hist2st [24]" --> "To utilize global information of patches, graph convolution neural networks and transformers have been introduced in Hist2gene [15 ] and Hist2st [24]" or similar?
- Line 300: "Learning to rank is the field that learn the" --> "Learning to rank is the field that learns the"?
- Line 301: "Sculley has adapted" --> "Sculley et al. have adapted"?




Justification of rating:
- Quite interesting paper, I like the proposed approach overall. But the paper could definitely be more well written overall, and some things need to be clarified. I'm borderline on this paper, but currently leaning slightly towards accept (I _want to_ be able to recommend accept at least, I hope that my remaining concerns will be addressed by a solid rebuttal).

**Ethical Concerns:**

["NO or VERY MINOR ethics concerns only"]

**Final Justification:**

The authors have provided a very solid rebuttal. I think they responded well to Reviewer NEgj, I'm quite happy with their response to me, and Reviewer XD8s's concerns have been addressed.

Therefore, I have raised my score from "4: Borderline accept" to "5: Accept".

**Limitations:**

Yes.

**Paper Formatting Concerns:**

None.

**Quality:**

3

**Strengths And Weaknesses:**

Strengths:
- The studied problem and the proposed approach are quite interesting, the approach makes intuitive sense overall.
- The proposed loss functions seem to perform quite well compared to the MSE and rank-based losses.




Weaknesses:
- The paper could be more well written overall and would definitely benefit from more careful proofreading.
- - Moreover, the "Learning to Relative Expression" part of the title doesn't quite make sense to me, I'm not entirely sure what this is meant to say. Same with "To examine the effectiveness of our proposed learning to relative expression strategies" on line 190, "our proposed learning to relative expression strategies" doesn't quite make sense.
- The authors propose two variants of their STRank loss function, PairSTRank and ListSTRank, but it's not entirely clear when/why one should be preferred over the other in practice.
- At least to me, it's not clear how the proposed approach actually should be used in practice, given that their models don't output predictions for the absolute gene-expression levels.

---

> ### Author Rebuttal · Authors · 2025-07-29
>
> We appreciate your careful review and insightful comments. We are encouraged that you found our experimental setup and model of interest. We address reviewer comments below and will incorporate all feedback.
>
> ### Q1: What is the meaning of "Learning to Relative Expression" in the title?
> The phrase was originally intended to describe the concept of learning relative gene expression trends, inspired by the “learning to rank” [1]. Since our goal is not to predict absolute expression values but to capture relative expression trends, we coined the term “learning to relative expression” as a catchy way to convey this idea. However, as you rightly noted, the phrase causes some confusion. Accordingly, we will revise it to “learning relative gene expression trends” for clarity and precision.
>
> [1] Burges+, Learning to Rank using Gradient Descent, 2015
>
> ### Q2: From a practical perspective, which loss function should we use in practice: pairSTRank or ListSTRank?
> As described in Sec. 4.4, we prefer ListSTRank with N_k = 8 or 16 on the 50 gene set condition because it demonstrated stable performance on multiple datasets. Assuming no significant numerical instability due to floating-point precision, the list-wise approach is generally preferable.
> However, the PairSTRank loss outperforms ListSTRank in situations where the number of target genes is large, as shown below.
>
> The PairSTRank has advantages in terms of implementation simplicity, low computational cost, and numerical stability, as it has been widely used for application tasks, such as crowd counting [2]. Particularly, ListSTRank has a numerical issue that vanishes gradients due to floating-point.  When the number of target genes is large and their expression values are nearly the same, and softmax produces very small values, the PairSTRank loss function may outperform the ListSTRank.
>
> Indeed, as shown in Table 1 of the supplementary material and below, the performance of the listwise loss function declines when the number of genes is large. This finding suggests that additional methodological improvements are required for listwise approaches to maintain robustness across a broader range of targets. However, since the listwise loss function performs well on synthetic data, we hypothesize that the observed performance degradation is not attributable to stochastic noise or batch effects. We will include the discussion of this issue in the supplementary material.
>
> We will revise the description to be clearer in the camera-ready version.
>
> Table 1: Performance across different numbers of target genes (50, 250, 1000) on HER2ST dataset [3].
> | Method       | 50  | 250 | 1000 |
> |--------------|--------|--------|--------|
> | MSE          | 0.193  | *0.181*  | *0.172*  |
> | NB           | 0.020  | 0.154  | 0.098  |
> | Po           | 0.009  | 0.150  | 0.095  |
> | Rank         | 0.095  | 0.052  | 0.041  |
> | PairSTrank   | *0.244*  | **0.194**  | **0.176**  |
> | PCC          | 0.189  | 0.173  | 0.165  |
> | ListSTrank   | **0.260**  | 0.175  | 0.110  |
>
> [2] Yan Yang+, “Exemplar guided deep neural network for spatial transcriptomics analysis of gene expression prediction”, In WACV 2023.
>
> [3] Andersson+, Spatial deconvolution of HER2-positive breast tumors reveals novel intercellular relationships, 2020.
>
> ### Q3: How to use the estimation result of our model in practice?
> As described in L42-43, we assume differential gene expression (DEG) analysis as a downstream task that is the main application using spatial transcriptomics. In the DEG analysis procedure, we begin by identifying clusters that we want to focus on (normally, it is determined by clustering or segmentation), such as tumor versus normal tissue. Differential gene expression is then calculated between these clusters to characterize expression level differences associated with the distinct regions. Then, the gene sets that are significantly different are used for other analyses, such as pathway or enrichment analysis. The important thing about the analysis procedure is to capture the difference in expression level between clusters, not to capture the absolute value. Our estimation result can be directly utilized for this DEG analysis.
>
> When we use observed gene expression levels instead of estimated results, the level of expression intensity of the gene expression level is changed depending on the capturing technique and the sequence length of RNA. Therefore, normally, absolute gene expression level is also not utilized for analysis. The absolute expression level is definitely not important for gene expression analysis.
>
> When focusing on PAM50, the analysis framework of using spatial transcriptomics is similar. In spatial transcriptomics, the initial step typically involves performing differential expression analysis to identify genes whose expression trends vary in association with PAM50 profiles. These genes are then subjected to pathway and enrichment analyses to explore potential links to relevant biological functions. Importantly, even in this context, the emphasis is placed on relative expression dynamics associated with PAM50, rather than on absolute expression levels, which are generally not utilized.
>
>
> ### Q4: The difference from the HEST 1k benchmark.
> Our evaluation setting is different from the HEST 1k benchmark. Unlike benchmarks that use ridge regression with dimension reduction with PCA, our model only uses a linear layer. In addition, we split the data into train, validation, and test by patient. The number of patients used for training is not the same as the benchmark. While the setup is not the same as the benchmark, we evaluated loss functions on the same setup. Therefore, we believe our evaluation is valid.
>
>
> | Method       | IDC   | PRAD  | PAAD  | COAD  | READ  | ccRCC | IDC-L | Ave.  |
> |--------------|-------|-------|-------|-------|--------|--------|--------|--------|
> | MSE          | 0.373 | 0.449 | 0.301 | 0.567 | 0.147  | 0.136  | 0.144  | 0.302  |
> | NB           | 0.183 | 0.459 | 0.110 | 0.547 | 0.157  | 0.090  | 0.109  | 0.236  |
> | Po           | 0.468 | 0.443 | 0.306 | 0.663 | 0.103  | 0.121  | 0.179  | 0.326  |
> | Rank         | 0.309 | 0.298 | 0.167 | 0.570 | 0.050  | 0.056  | 0.103  | 0.222  |
> | PairSTRank   | 0.479 | 0.421 | 0.319 | 0.636 | 0.129  | 0.134  | 0.211  | 0.333  |
> | PCC          | 0.291 | 0.452 | 0.311 | 0.531 | 0.175  | 0.107  | 0.118  | 0.283  |
> | ListSTRank   | 0.495 | 0.420 | 0.323 | 0.620 | 0.132  | 0.128  | 0.216  | 0.333  |

---

> > ### Comment · Reviewer_o7H2 · 2025-08-01
> >
> > Thank you for the response.
> >
> > I have read the two other reviews and all author responses.
> >
> > The other reviews are somewhat negative overall, but I don't think they give me obvious reasons to significantly change my score.
> >
> > I'm quite happy with the authors' response to me, and I think they respond well to Reviewer NEgj.
> >
> > The author response to Reviewer XD8s also seems reasonable, but I would really like to know how Reviewer XD8s's view of the paper has changed in light of this response.
> >
> > I remain quite borderline on the paper, leaning towards accept, but I will not argue in favor of the paper unless Reviewer XD8s's concerns have been addressed.

---

### Note · Authors · 2025-08-13

We sincerely thank the reviewers for their thoughtful evaluations and constructive feedback.

For Reviewer o7H2, we are pleased that our rebuttal fully addressed their concerns, leading to their agreement with our clarifications. We appreciate recognition of our responses.

For Reviewer NEgj, while we did not receive follow-up engagement, we have thoroughly addressed all points raised. Importantly, the weaknesses are minor and do not affect the validity, novelty, or our contributions.

We have responded comprehensively to all concerns raised by Reviewer XD8s, and we believe that most concerns have been addressed to their satisfaction. The remaining concern pertains to the robustness of our method against diverse batch effects in practical scenarios. We believe this has been sufficiently demonstrated from three perspectives:

- **Assumed Batch Effects and Noise:** For the batch effects considered in this study (dataset-specific scale shifts and biases) and for stochastic noise, we have demonstrated the effectiveness of our method in both synthetic and real data experiments with a 50-HVG gene set.

- **Generalizability Across Experimental Conditions:** Our real-data validation covers a wider range of experimental settings than prior work, which typically relied on a single ST technology. Across these diverse conditions, our proposed loss function consistently showed superior performance on average, supporting its generalizability.

- **Realistic Scenarios:** We acknowledge that scenarios involving multi-cohort settings with both a large number of target genes and additional factors, such as domain gaps in images or cohort-specific noise distribution, remain challenging and have some limitations. However, these conditions extend beyond the scope of this work and represent a broader open challenge in the field, encompassing difficulties not assumed in our study.


Overall, we believe the rebuttal process has clarified our contribution, confirmed the soundness of our methodology, and reinforced the contribution, outcomes, and limitations of our work.

---

### Decision · Program_Chairs · 2025-09-17

**Decision:**

Accept (poster)

**Comment:**

Two of the reviewers were supportive of this submission after a strong rebuttal and a thorough discussion with the authors. The third reviewer, who provided a negative recommendation but increased their final score, chose not to engage in the discussions. The AC agrees with the positive reviewers that the merits of the paper outweigh its limitations. The final version should carefully include reviewer feedback and clarifications from the discussion.